# Principles of dengue virus evolvability derived from genotype-fitness maps in human and mosquito cells

**Patrick T Dolan[1,2†], Shuhei Taguwa[1†], Mauricio Aguilar Rangel[1], Ashley Acevedo[2], Tzachi Hagai[3], Raul Andino[2]\*, Judith Frydman[1]\***

[1]Stanford University, Department of Biology, Stanford, United States; [2]University of California, Microbiology and Immunology, San Francisco, San Francisco, United States; [3]Shmunis School of Biomedicine and Cancer Research, George S. Wise Faculty of Life Sciences, Tel Aviv University, Tel Aviv, Israel

**Abstract** Dengue virus (DENV) cycles between mosquito and mammalian hosts. To examine how DENV populations adapt to these different host environments, we used serial passage in human and mosquito cell lines and estimated fitness effects for all single-nucleotide variants in these populations using ultra-deep sequencing. This allowed us to determine the contributions of beneficial and deleterious mutations to the collective fitness of the population. Our analysis revealed that the continuous influx of a large burden of deleterious mutations counterbalances the effect of rare, host-specific beneficial mutations to shape the path of adaptation. Beneficial mutations preferentially map to intrinsically disordered domains in the viral proteome and cluster to defined regions in the genome. These phenotypically redundant adaptive alleles may facilitate host-specific DENV adaptation. Importantly, the evolutionary constraints described in our simple system mirror trends observed across DENV and Zika strains, indicating it recapitulates key biophysical and biological constraints shaping long-term viral evolution.

\*For correspondence:
raul.andino@ucsf.edu (RA);
jfrydman@stanford.edu (JF)

[†]These authors contributed equally to this work

**Competing interests:** The authors declare that no competing interests exist.

## Introduction

The evolutionary capacity of RNA viruses allows them to rapidly adapt to their environment and overcome barriers to infection (*Alto et al., 2013*; *Dolan et al., 2018*; *Domingo and Perales, 2014*; *Sanjuán, 2016*). Quantifying the evolutionary dynamics of virus populations in controlled experimental systems can reveal biological constraints on the viral genome, molecular mechanisms of viral adaptation, and fundamental biophysical and population genetic principles governing molecular evolution in general.

Arthropod-borne viruses, or arboviruses, such as Dengue (DENV), Zika (ZIKV), and Chikungunya (CHIKV), are a significant cause of disease globally, with half of the world's population exposed to arboviral vectors. DENV alone causes approximately 390 million infections and 10,000 deaths annually (*Bhatt et al., 2013*; *Messina et al., 2019*). Arboviruses must cycle between vertebrate and invertebrate hosts, which differ significantly in body temperature, cellular environment, and mode of antiviral immunity, raising questions about the evolutionary strategies they may employ to replicate in these different host environments. Several studies have addressed these alternative landscapes experimentally in vitro and in vivo, identifying mutations that confer increased fitness in each host (*Coffey and Vignuzzi, 2011*; *Filomatori et al., 2017*; *Forrester et al., 2014*; *Greene et al., 2005*; *Pompon et al., 2017*; *Sessions et al., 2015*; *Stapleford et al., 2014*; *Villordo et al., 2015*). However, we still lack a comprehensive picture of the alternative genotype-fitness landscapes of any arbovirus defined by the human and insect host environments. Comparing the evolutionary dynamics

**eLife digest** Viruses are constantly evolving as a result of mutations in their genetic material and environmental pressures. Viruses switching between insects and mammals face unique evolutionary pressures because they must retain their ability to infect both types of organisms. Yet, the mutations in a virus that may be beneficial in an insect may be different from the ones that may be beneficial in a mammal. Mutations in one host may be even harmful in the other.

To learn more about how such viruses thrive as they switch between hosts, Dolan, Taguwa et al. studied the dengue virus, which causes over 390 million infections and over 10,000 deaths each year around the globe. They compared the mutations that occurred as the virus multiplied in human and mosquito cells grown in a laboratory.

In the experiments, they used a method called ultra-deep RNA sequencing to identify every change that occurred in the genetic material of the virus each time it multiplied. They determined whether the mutations were beneficial or harmful based on whether they became more common – suggesting they helped the virus survive – or whether they did not persist because they were likely harmful or even fatal to the virus.

The experiments showed that many harmful mutations constantly occur in the virus, in both human and mosquito cells. Beneficial changes happen rarely, and those that do are usually only helpful in one type of cell. Fatal mutations tended to occur in parts of the genetic material that encodes regions in the viral proteins that must remain the same. These structural elements appear to be essential to the virus's survival and unable to undergo change, which makes them good targets for antiviral drugs or vaccines. The techniques used in the study may be useful for investigating other viruses and for understanding the evolutionary constraints on viruses more generally. This may help scientists develop antiviral drugs or vaccines that will remain effective even as viruses continue to evolve and mutate.

---

of viral populations across different host environments could highlight key points of host-specific selection and define the patterns of evolutionary constraint that define the landscape in each host.

RNA viruses exist as a dynamic population of co-circulating genotypes surrounding a master sequence (*Domingo, 2002*; *Domingo et al., 2012*; *Holland et al., 1992*; *Lauring and Andino, 2010*; *Wilke, 2005*). It is becoming increasingly clear that the distribution and dynamics of minor alleles play important roles in population fitness, adaptation, and disease (*Andino and Domingo, 2015*; *Bordería et al., 2015*; *Grad et al., 2014*; *Shirogane et al., 2012*; *Vignuzzi et al., 2006*; *Xue et al., 2018*, *Xue et al., 2016*); furthermore, the neighborhood of connected genotypes is thought to be important to population fitness (*Moratorio et al., 2017*). The genomes of viruses that alternate between hosts, such as arboviruses, are subject to selection in distinct environments, raising the question of how these viruses maintain fitness over alternating hosts.

Emerging deep-sequencing techniques allow us to probe the mutational landscape. Library-based methods, such as deep mutational scanning (DMS), screen defined collections of sequences against specific selective pressures. DMS can quantify the fitness effects of individual mutations in viral genomes through intentional diversification of protein sequences (*Ashenberg et al., 2017*; *Setoh et al., 2019*; *Thyagarajan and Bloom, 2014*; *Visher et al., 2016*). However, these approaches do not capture the evolutionary dynamics of natural populations. The analysis of naturally occurring variation and evolution in experimental virus populations has been limited to allele frequencies greater 1 in 1000, due to the error rates associated with reverse transcriptase used in cDNA synthesis. Recently, high-accuracy sequencing approaches that control for sequencing errors through barcoding, like PrimerID (*Jabara et al., 2011*), or through template circularization and amplification, like Circular Sequencing (CirSeq) (*Acevedo et al., 2014*; *Acevedo and Andino, 2014*), can detect alleles as rare as 1 in $10^6$ in frequency. This sequencing depth enables the observation of the full spectrum of diversity in samples from evolving viral populations. Thus, the ability to globally trace the evolutionary dynamics of individual alleles in viral populations from their genesis at the mutation rate to their eventual fate in a given experiment allows us to describe the viral fitness landscapes that shape adapting populations in unprecedented detail. Importantly, this depth allows observation of common variants that accumulate under positive selection in each experiment, but

also reveals rare variants limited to low frequencies by negative selection. This permits quantification of the contribution of genetic constraint on the viral adaptation process.

We here use CirSeq to characterize the fitness landscapes of DENV populations adapting to the distinct environments and cellular machineries of human and mosquito cells by tracing individual allele trajectories for almost all possible single nucleotide variants across the DENV genome. This analysis reveals the influence of both positive and negative selection in shaping the evolutionary paths of DENV in these distinct cellular environments. Analysis of the allele repertoire reveals how fitness of the viral population represents a balance between the dynamics of rare beneficial mutations and the significant and constantly replenished load of deleterious alleles during adaptation. We find that adaptation relies on host-specific beneficial mutations that are clustered in specific regions of the DENV genome and enriched in regions of the proteome that exhibit structural flexibility. Of note, these regions are also sites of variation across naturally occurring DENV and ZIKV strains indicating that our analysis provides insights into genetic and biophysical principles of flaviviral evolution and reveals parallels between long- and short-term evolutionary scales.

## Results

### Phenotypic characterization of DENV populations adapting to human or mosquito cells

Two simple models could describe how arboviruses cycle between their alternative host environments (*Figure 1a*). First, the viral genome could have overlapping host-specific fitness landscapes; in this case, transmission would not involve significant trade-offs. Alternatively, the virus may have distinct host-specific landscapes with offset fitness maxima. To characterize the relative topography of the adaptive landscapes of DENV in vertebrate and invertebrate hosts (*Figure 1a*), we experimentally evolved DENV in human and mosquito cell lines (*Figure 1b*). Starting from infectious vRNA transcribed from a plasmid encoding dengue virus type 2 (DENV Type 2, Thailand/16681/84), we serially passaged viral populations in two well characterized cell lines used for DENV research: the human hepatoma-derived cell line Huh7 or the *Aedes albopictus*-derived cell line C6/36, for nine passages. Although *Ae. aegypti* is the primary mosquito vector of DENV, *Ae. albopictus* is increasingly understood to be an urban vector species in DENV transmission (*Kamgang et al., 2019*; *Lambrechts et al., 2010*; *Moncayo et al., 2004*; *Rezza, 2012*).

To control the influence of drift due to genetic bottlenecks and recombination and complementation between viral variants (*Clarke et al., 1993*; *Wahl et al., 2002*), each passage infected $5 \times 10^6$ cells at a multiplicity of infection (MOI) of 0.1, using a viral inoculum of $5 \times 10^5$ focus forming units (FFU) from the previous passage. We estimate the virus undergoes 1–3 rounds of replication in each passage. To distinguish host-specific versus replicate-specific events, we passaged two lineages in parallel experiments in each cell line following transfection into each cell line (Series A and B, *Figure 1b*).

The fitness gains associated with adaptation were assessed phenotypically by measurements of virus titer (*Figure 1c* and *Figure 1—figure supplement 1a*, *Figure 1—source data 1*), intracellular vRNA content (*Figure 1d* and *Figure 1—figure supplement 1b*, *Figure 1—source data 2*), and focus size and morphology (*Figure 1e* and *Figure 1—figure supplement 1c and d*) for each viral population in the passaged host cell. All of these fitness measures increased over time for the passaged host, indicating significant adaptive evolution throughout the experiment. We quantified fitness trade-offs in parallel by carrying out the same measurements in the alternative (by-passed) host cell. In agreement with previous studies (*Byk and Gamarnik, 2016*; *Greene et al., 2005*; *Johnson et al., 1994*; *Novella et al., 1995*; *Vasilakis et al., 2009*; *Villordo et al., 2015*; *Villordo and Gamarnik, 2013*), passaging on one host cell line was accompanied by a concurrent loss of fitness in the alternative host cell line (*Figure 1c* and *Figure 1—figure supplement 1e*). For instance, the human-adapted virus showed a uniform small focus phenotype when plated on mosquito cells (*Figure 1f*). In contrast, mosquito-adapted populations formed fewer foci in human cells (*Figure 1f*). Mosquito-adapted populations exhibited a heterogeneous focus phenotype, with small and large foci, suggesting they contain distinct variants which differentially affect replication in human cells.

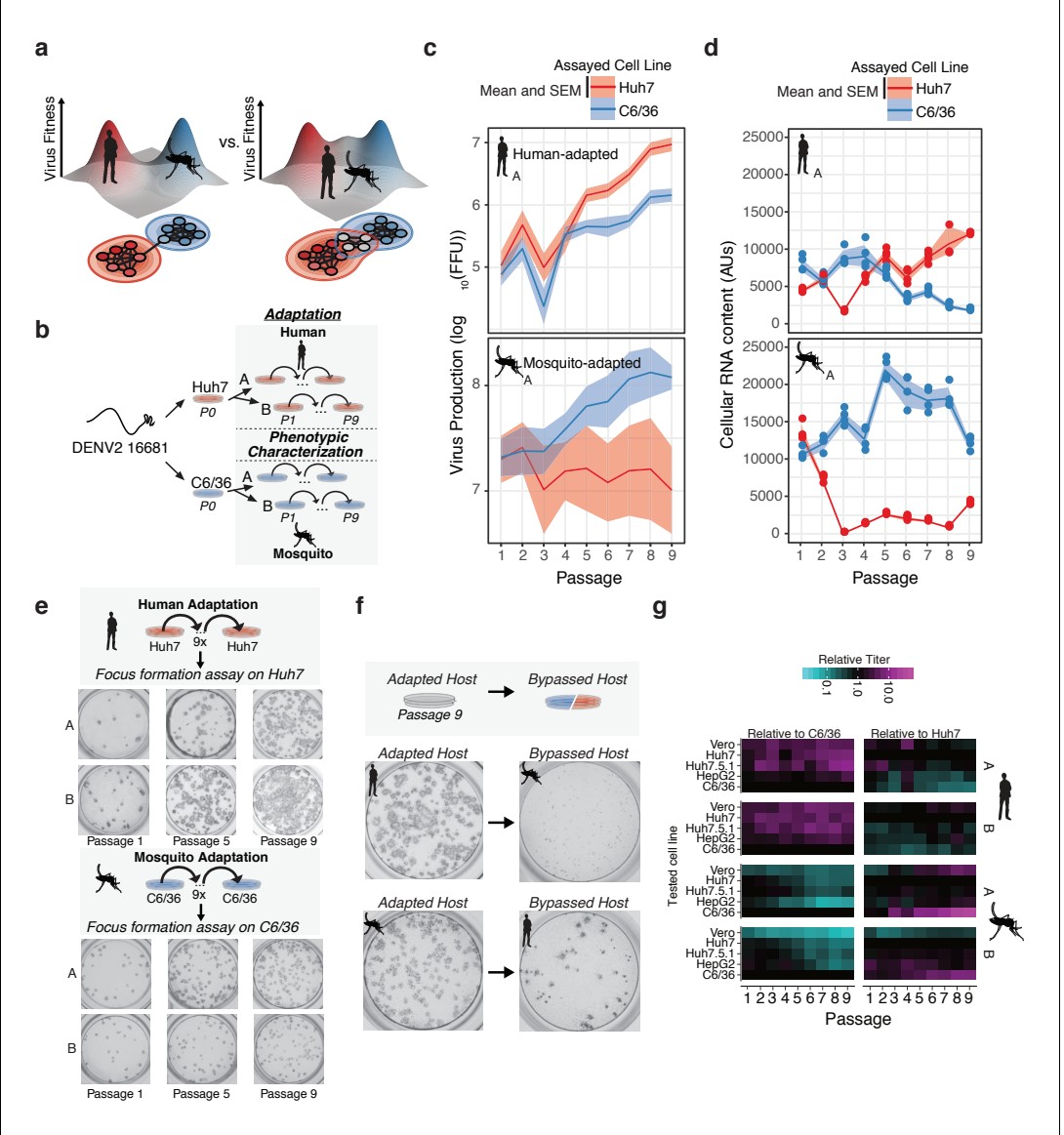

**Figure 1.** Dengue navigates distinct fitness landscapes in its alternative hosts. (**a**) Two potential models of the genotype-fitness landscape and mutational network in alternative arboviral hosts. The relative topography of the viral genotype-fitness landscape determines the extent of evolutionary trade-offs associated with transmission, and the paths through the mutational network toward host adaptation, and the proportion of genotypes viable in the alternative host environment (gray nodes). (**b**) Outline of our in vitro DENV evolution experiment. Dengue virus RNA (Serotype 2/16881/Thailand/1985) was electroporated into mosquito (C6/36) or human cell lines (Huh7), and the resulting viral stocks were passaged at fixed population size (MOI = 0.1, $5 \times 10^5$ FFU/passage) for nine passages in biological duplicates. After passage, samples of virus from each passaged population were characterized for phenotypic measures of fitness. (**c**) Viral production assays comparing mosquito-adapted (top panel) and human-adapted (bottom panel) DENV populations. Adapted populations show increased virus production on their adapted hosts. Biological replicate A is shown for all experiments. (**d**) Analysis of viral RNA content by qRT-PCR. Cellular DENV RNA is significantly decreased when adapted lines are propagated on the by-passed, alternative host. Lines and shading represent the mean and standard deviation of four technical replicates, respectively. Biological replicate A is shown for all experiments. (**e**) Focus forming assays of the adapted lineages performed on each population over the course of passage. Passages 1, 5, and 9 are shown for all lineages. Focus size increased markedly throughout passage on the adapted host. (**f**) Focus forming assays of the P9 virus on the adapted (left) and by-passed (right) host. Changes in focus size and morphology suggest evolutionary trade-offs between the alternative hosts. (**g**) Heatmap showing the virus production of the passaged lineages in human, primate, and mosquito cell lines.

*Figure 1 continued on next page*

*Figure 1 continued*

The online version of this article includes the following source data and figure supplement(s) for figure 1:

**Source data 1.** Virus titer data from focus forming assays on multiple cell lines.
**Source data 2.** Table of intracellular RNA content measurements.
**Figure supplement 1.** Phenotypic characterization of passaged viral populations.

We further assessed the evolutionary trade-offs during host adaptation by comparing the relative titers of all the passaged populations in both the original and the alternative host cells, Huh7 and C6/36. To examine if the fitness effects were specific to the Huh7 cell line used or reflected a broader (de-)adaptation to the mammalian cell environment, we also measure fitness in two additional human cell lines, Huh7.5.1 cells, human hepatoma-derived HepG2 cells, as well as one African Green Monkey epithelial-derived cell line, Vero (*Figure 1g* and *Figure 1—figure supplement 1e*). For each passage, viral titers were normalized to that obtained in the adapted (original) host cell line, to yield the efficiency of plating (EOP) (individual EOP plots shown in *Figure 1—figure supplement 1d* and as a heatmap in *Figure 1g*). Similar EOPs were observed for all primate-derived cells, indicating the adaptation and de-adaptation observed upon passage in either Huh7 or C6/36 cells largely reflect changes in fitness to the mammalian vs insect cell environments and distinct cellular machineries. Of note, cultured cells are often deficient in some innate immune pathways. For instance, C6/36 cells exhibit altered RNA-mediated antiviral immunity (*Brackney et al., 2010*; *Scott et al., 2010*). Huh7.5.1 cells are RIG-I-deficient (*Shirasago et al., 2015*; *Zhong et al., 2006*) while Vero cells are deficient in type-I interferon production (*Saito et al., 2020*; *Desmyter et al., 1968*; *Osada et al., 2014*). Intriguingly, viral populations exhibit an intermediate phenotype in Huh7.5.1 cells relative to the other primate and insect cell lines, which are distinct from their phenotype in Vero cells. In the future, it will be interesting to extend these analyses to intact infected hosts to clarify how innate and organismal immunity contribute to host-specific adaptation and host tropism.

## Characterizing genotypic changes in adapting DENV populations

To determine the genotypic changes associated with host cell adaptation, we subjected all viral populations to CirSeq RNA sequencing (*Acevedo and Andino, 2014*; *Whitfield and Andino, 2016*). CirSeq achieves error-correction through an experimental-computational innovation wherein consensus sequences are derived from individual template RNAs. By fragmenting and circularizing the viral template RNA and generating circular reverse transcripts, the CirSeq pipeline computationally determines the corrected consensus sequence through alignment of the concatenated sequences in each individual short read. With an error rate of less than 1 in $10^6$, CirSeq yielded an average of approximately $2 \times 10^5$–$2 \times 10^6$ reads per base across the genome for each viral population in our experiments (*Figure 2a*; *Acevedo et al., 2014*; *Whitfield and Andino, 2016*). This depth permits the accurate quantification of alleles as rare as 1 in 60,000–600,000 genomes (*Figure 2b*).

We next examined the allele frequencies in each passage for each position of the DENV genome (*Video 1*, Passage 7 shown in *Figure 2b*). Most alleles are present at low frequencies, between 1 in 1000 and 1 in 100,000. However, many alleles rapidly increased in frequency with passage number (*Video 1*). No mutations reached fixation over the course of 9 passages, with the highest allele frequencies reached near 80% by the end of the experiment. This may reflect the role of clonal interference in the evolution dynamics of complex populations. Comparing the allele frequencies in the two independent passage series A and B revealed that, as passage number increased, the replicate populations (*Figure 2c* i or ii) shared numerous high-frequency mutations (defined here as >1% allele frequency) while populations passaged in different hosts shared no high frequency mutations (*Figure 2c* iii, *Figure 2—figure supplement 1a–b*). As evident in *Video 1*, cell-adaptation increased the frequency of alleles in specific regions of the DENV genome, such as NS2A and NS4B in human-adapted lineages and in E, NS3, and the 3′ UTR in mosquito-adapted replicates (*Figure 2c*).

To better visualize the high-dimensional temporal dynamics of adaptation (*Figure 2—figure supplement 1a–b*, *Video 1*), we employed two alternative dimension reduction approaches, principal components analysis (PCA) (*Figure 2d* and *Figure 2—figure supplement 1a and b*) and multidimensional scaling (MDS) to analyze the population sequencing data.

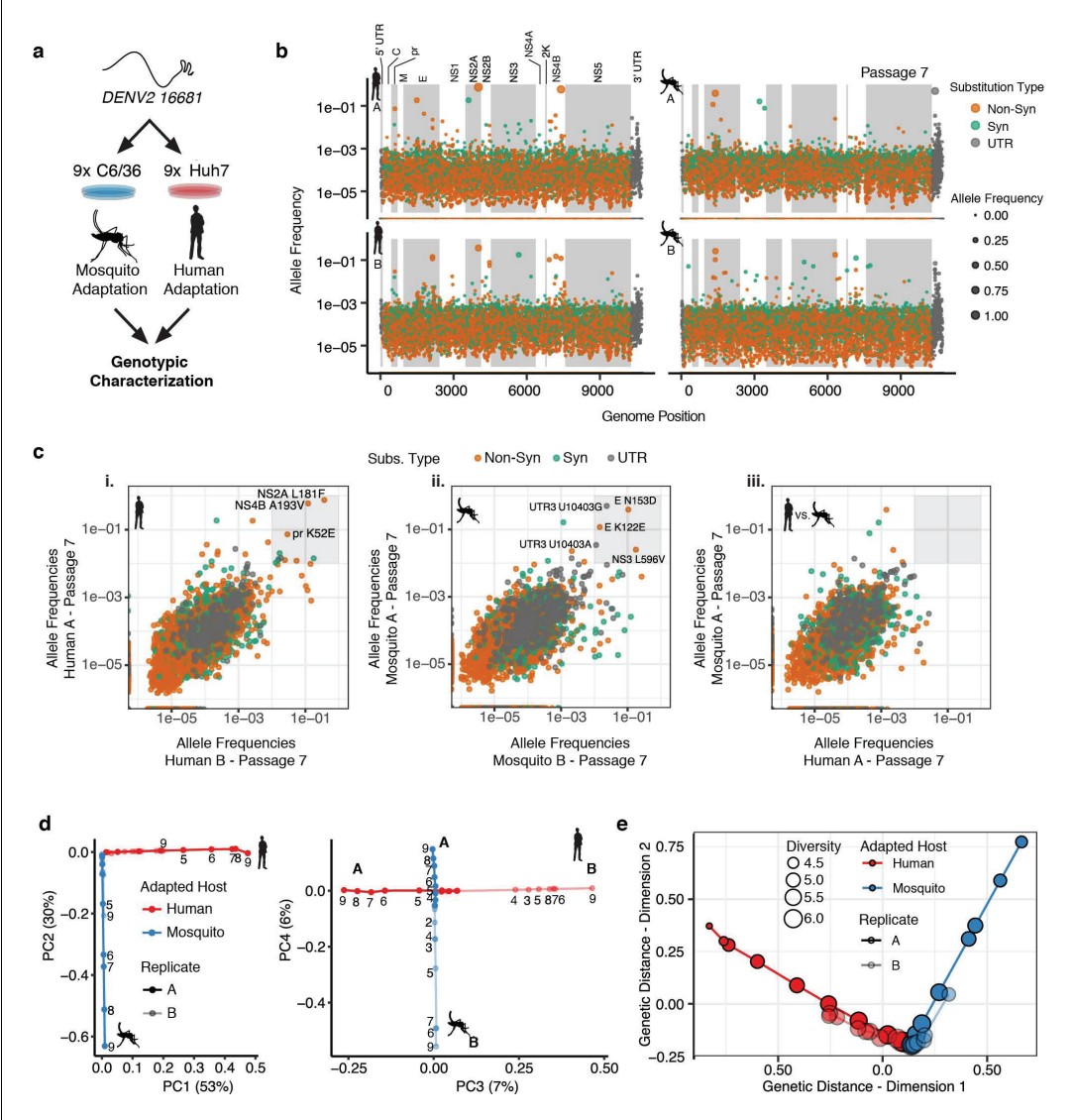

**Figure 2.** Adapting viral lineages show host-specific patterns of genetic variance. (**a**) Adapted viral populations were subject to genotypic characterization by ultra-deep sequencing using the CirSeq procedure. (**b**) Plots of allele frequency across the viral genomes for all four viral populations at passage 7. Alleles are colored by mutation type (Nonsynonymous, Orange; Synonymous, Green; Mutations in the untranslated region (UTR), Dark Gray). Shaded regions denote mature peptide boundaries in viral ORF. (**c**) Scatter plots comparing allele frequencies between adapted populations of human- and mosquito-adapted dengue virus. Replicate host-adapted populations share multiple high-frequency non-synonymous mutations, but populations from alternative hosts do not (gray square, >10%). (**d**) Dimension reduction of the allele frequencies by principal components analysis summarizes the host-specific patterns of variance (left), and the replicate-specific differences in genetic variability over passage (right). (**e**) A two-dimensional embedding of the pairwise genetic distances between the sequenced viral populations (Weir-Reynolds Distance) by multidimensional scaling. The viral populations (red- and blue-hued trajectories) project out from the founding genotype in orthogonal and host-specific directions.

The online version of this article includes the following figure supplement(s) for figure 2:

**Figure supplement 1.** Genotypic characterization of passaged DENV2 populations.

PCA quantifies the common patterns of allele frequency variance between the populations, identifying independent patterns of variance. The first four components of the PCA explained 96% of the observed allele frequency variance in the experimental populations. The first two components, which explain 83% of the observed variance (*Figure 2d left panel* and *Figure 2—figure supplement 1c*), partitioned the viral lineages along two orthogonal, host-specific paths, radiating outward in order of passage number from the original WT genotype (*Figure 2d, left panel*). The third and fourth

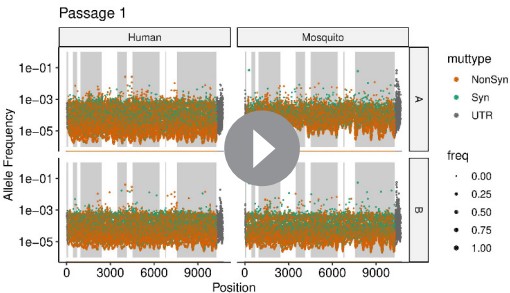

**Video 1.** Animation of the allele frequencies in the adapting populations over nine passages. Colors: Orange, non-synonymous mutations; Green, synonymous mutation; and Gray, mutations in the UTR. https://elifesciences.org/articles/61921#video1

components in the PCA explained 13.4% of the observed variance and further partitioned each lineage along orthogonal, replicate-specific axes (***Figure 2d right panel*** for human series A and B and mosquito series A and B). PCA-derived scores for individual alleles in component space summarized their contribution to the host- and replicate-specific dynamics (***Figure 2—figure supplement 1d***). The 3′ UTR and E contained the strongest signatures of mosquito-specific adaptation. Human-specific alleles were distributed across the genome, including nonsynonymous substitutions in E, NS2A, and NS4B (***Figure 2—figure supplement 1d***). Replicate-specific PCA components also highlighted clusters of alternative alleles in E and the 3′ UTR. These analyses revealed the contribution of host-specific and replicate-specific changes in the viral population.

Multidimensional scaling (MDS), which allows the embedding of the multidimensional pairwise genetic distances between populations into two dimensions (***Figure 2e***), provides a complementary view of the genetic divergence of the populations. MDS also revealed the orthogonal, host-specific evolutionary paths of the populations (***Figure 2e***). The finding that reproducible population structures emerge during DENV adaptation to each host resonates with theoretical predictions that large viral populations will develop genetic structures, often called quasispecies, deterministically based on the selective environment (***Holland et al., 1992***; ***Lauring and Andino, 2010***; ***Sardanyés et al., 2008***; ***Wilke, 2005***). The host-specific composition of these populations likely reflects the differences in the selective environments that determine host range and specificity, prompting us to dissect their composition further.

## Fitness landscapes of DENV adaptation to human and mosquito cells

The concept of fitness links the frequency dynamics of individual alleles in a population with their phenotypic outcome, that is, beneficial, deleterious, lethal, or neutral. Lethal and deleterious alleles are held to low frequencies by negative selection, while beneficial mutations increase in frequency due to positive selection (***Figure 3a***). Observing the frequency trajectory of a given allele over time relative to its mutation rate enables the estimation of its fitness effect.

The high accuracy of CirSeq allowed us to estimate the substitution-specific per-site mutation rates for DENV using a previously described maximum likelihood (ML) approach (***Figure 3—figure supplement 1a***; ***Acevedo et al., 2014***). These estimates, ranging between $10^{-5}$ and $10^{-6}$ substitutions per nucleotide per replication (s/n/r) for each substitution, agreed well across populations. C-to-U mutations occurred at the highest rate, approximately $5 \times 10^{-4}$ s/n/r in all populations. This higher C-to-U mutation rate may reflect poor base selection by the polymerase, spontaneous deamination of the template RNA, or the action of cellular deaminases such as APOBEC3 enzymes (***Milewska et al., 2018***; ***Sanjuán, 2016***; ***Milewska et al., 2018***; ***Sanjuán, 2016***). The genomic mutation rate, substitutions per genome per replication (s/g/r) ($\mu_g$), was calculated by taking the sum of the ML mutation rate estimates of all single-nucleotide mutations across the genome, yielding $\mu_g$ estimates of 0.70 and 0.73 s/g/r for mosquito populations and 0.61 and 0.60 s/g/r for human populations (***Supplementary file 2***). These estimates, indicating that the virus has a probability of acquiring less than one mutation per genome per replication cycle (***Figure 3f***), are consistent with genomic mutation rate estimates for other positive-strand RNA viruses (***Drake and Holland, 1999***).

Using a model derived from classical population genetics (***Figure 3b***), we next generated point estimates and 95% confidence intervals of relative fitness ($w$) for each possible allele in the DENV genome (***Figure 3c and d***). The distribution of mutational fitness effects, or DMFE, is commonly used to describe the mutational robustness of a given genome (***Figure 3c*** and ***Figure 3—figure supplement 1b***; ***Carrasco et al., 2007***; ***Sanjuán et al., 2004***; ***Visher et al., 2016***). Importantly, describing the full DMFE requires resolving the fitness effects of the large fraction of alleles with deleterious fitness effects. This requires significant sequencing depth to establish the behavior of these alleles

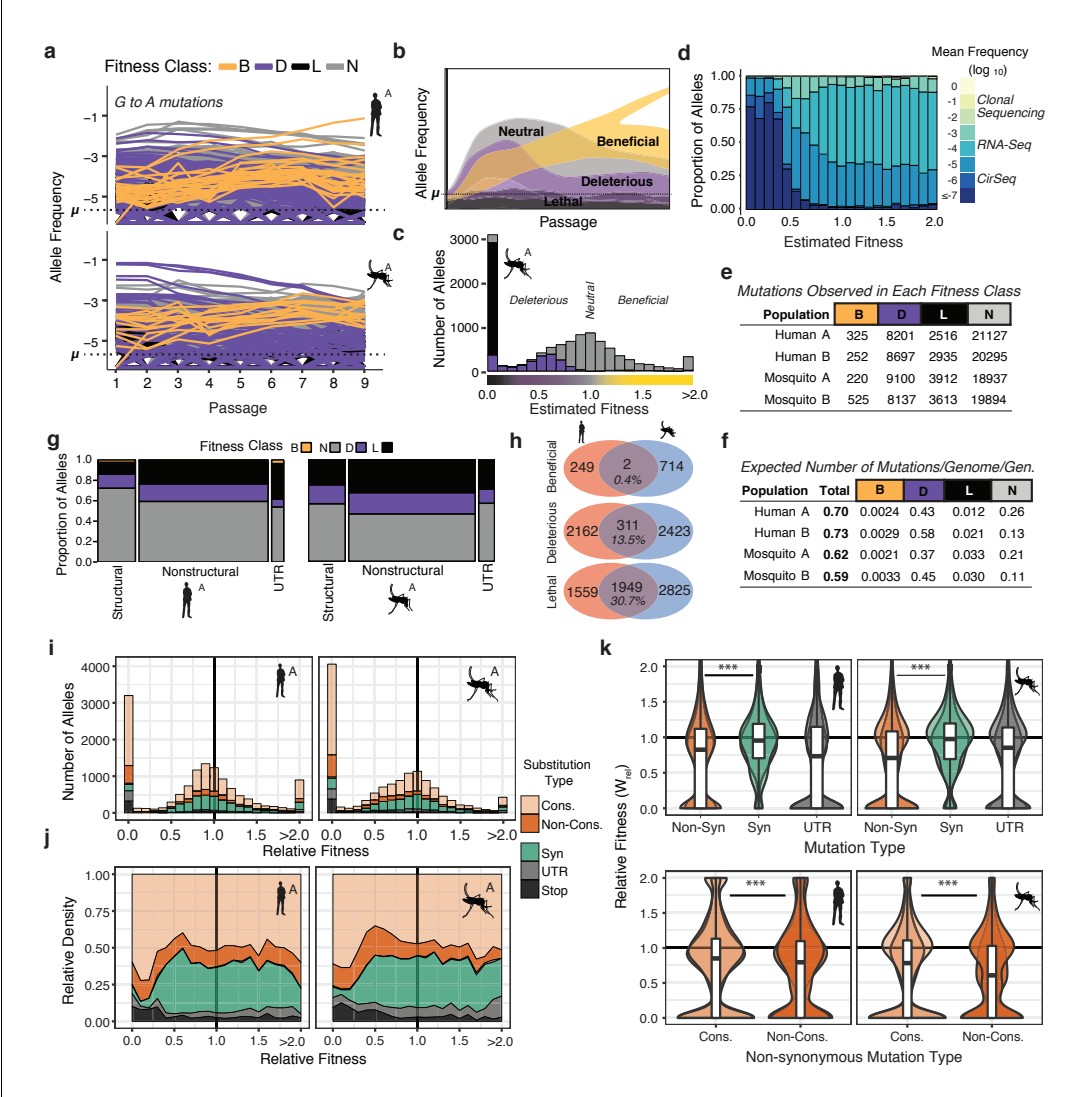

**Figure 3.** The distribution of fitness effects reveals patterns of evolutionary constraint. (a) The frequency trajectories for G-to-A mutations in the adapting populations determined by CirSeq. Colors represent the classification of each allele as beneficial, deleterious, lethal, or neutral (not statistically distinguishable from neutral behavior) (b) Schematic illustrating the expected frequency behavior of specific fitness classes relative to their corresponding mutation rate, μ. Changes in allele frequency between passages are used to estimate the fitness effects of individual alleles in the population (see Materials and methods). (c) Histogram showing the distribution of mutational fitness effects (DMFE) of DENV passaged in mosquito cells. The data shown are from mosquito A and represent the high confidence set of alleles (see text). The fitness classifications of alleles in each bin, based on their 95% confidence intervals, is indicated by the fill color. (d) The relative density of each mutation type across the fitness spectrum illustrates the sequencing depth necessary to observe regions of the fitness spectrum. Fill color represents the average frequency of the mutation over passage. (e) Tabulation of all alleles by fitness class. (f) Estimate of the genomic mutation rate per genome per generation ('Total'), and fitness class-specific mutation rates ('B', 'D', 'N', and 'L', *Supplementary file 2*). (g) Area plot showing the fitness effects associated with mutations in structural, non-structural, and UTR regions of the DENV genome. The relative width of the columns indicates the number of alleles in each class, the relative height of the colored regions indicates the proportion of alleles of a given class. (h) Venn diagrams showing the number of mutations identified as beneficial, deleterious, or lethal in the high confidence set alleles (see Materials and methods). These counts include alleles identified in either A or B replica populations. (i) Histograms of the DMFE broken down by mutation type. (j) Density plot of the relative density of mutation types across the DMFE to emphasize the local enrichment of specific classes. (k) Violin plots showing the relative fitness of nonsynonymous and synonymous mutations, and those in the viral UTRs. Overlapping plots are shown for replicates A and B for each host. Boxplots are computed based on the DMFE of both replica lineages in each host. Nonsynonymous mutations can further be partitioned into conservative and non-conservative classes.

The online version of this article includes the following figure supplement(s) for figure 3:

**Figure supplement 1.** Details of mutation rates, distributions of mutational fitness effects, and fitness class assignments.

relative to the mutation rate (*Figure 3b*). The vast majority of alleles in viral populations cannot be detected by clonal sequencing or conventional deep-sequencing approaches (*Figure 3d* and *Figure 3—figure supplement 1c*), which can only detect a few high-frequency beneficial and neutral mutations. Due to its low error rate, CirSeq enables the analysis of low-frequency alleles illuminating the contribution of deleterious and neutral mutations to the topography of the fitness landscape (*Figure 3d*).

The DMFEs of DENV exhibited bimodal distributions with peaks at lethality ($w=0$) and neutrality ($w=1.0$), and a long tail of rare beneficial mutations ($w>1.0$), similar to what is observed for other RNA viruses (*Acevedo et al., 2014*; *Minicka et al., 2017*; *Sanjuán et al., 2004*; *Visher et al., 2016*). The 95% CIs of these $w$ fitness estimates were used to classify individual alleles as beneficial (*B*), deleterious (*D*), lethal (*L*), or neutral (*N*) (*Figure 3e* and *Figure 3—figure supplement 1a*). Alleles with fitness 95% CI maxima equal to 0 were classified as lethal alleles; these never accumulate above their mutation rate due to rapid removal by negative selection (*Figure 3b and c*, black). Alleles with an upper CI higher than 0 but lower than 1.0 were considered deleterious (*Figure 3b* and *Figure 3—figure supplement 1b*, purple). Alleles with a lower CI greater than 1.0 were classified as beneficial; these accumulate at a rate greater than their mutation rate due to positive selection (*Figure 3b* and *Figure 3—figure supplement 1b*, yellow). Alleles whose trajectories could not be statistically distinguished from neutral behavior ($w=1.0$) are referred to as 'neutral' (*Figure 3b and e*, and *Figure 3—figure supplement 1b*, gray). Based on these classifications, we assessed the total proportion of mutations in each class, finding 8–12% of variants are lethal, 25–28% significantly deleterious, and only 0.5–1.5% significantly beneficial (*Figure 3e*).

The genomic mutation rate represents the rate at which novel mutations enter the population (*Figure 3f*, 'Total'). To understand the expected fitness of new mutations, we used fitness classifications for all 32,166 possible single-nucleotide variant alleles (*Figure 3e*) to estimate the genomic beneficial, deleterious, and lethal mutation rates (*Figure 3f*, *Supplementary File 2*). These estimates indicate that the virus maintains a substantial deleterious genetic load due to the high rate at which deleterious and lethal mutations flow into the population. We estimate DENV genomes have a 40–50% probability to acquire a deleterious mutation but only a 0.2–0.3% probability to acquire a beneficial mutation per replication cycle (*Figure 3f*).

## Defining constraints shaping the DENV fitness landscape

We next determined the proportion of mutations in each fitness class mapping to structural and non-structural regions of the viral polyprotein. A high confidence set of 13–14,000 alleles in each population was chosen based on sequencing depth and quality of the fit in the $w$ fitness estimates across passages. There were striking differences in the distribution of lethal and deleterious mutations in distinct regions of the genome (*Figure 3g*). Non-structural proteins were significantly enriched in deleterious and lethal mutations compared to structural proteins. This finding contrasts with results obtained from analyses of poliovirus population dynamics (*Acevedo et al., 2014*). Whereas DENV structural proteins exhibit higher mutational robustness compared to non-structural proteins, poliovirus structural proteins were found to be less robust to mutation than nonstructural proteins (*Acevedo et al., 2014*). Interestingly, in a mutational screening study in Influenza A, another enveloped virus, membrane-associated HA and NA proteins were more robust to mutation than 'internal' proteins (*Visher et al., 2016*). These differences likely arise from the distinct folding and stability constraints of the enveloped and non-enveloped virion structure of these different virus families. We also find the viral UTRs exhibit host-specific patterns of constraint, consistent with their host-specific roles in the viral life cycle (*Lodeiro et al., 2009*; *Villordo et al., 2015*, *Villordo et al., 2010*). In human cells, the DENV UTRs were more brittle but also contained more beneficial alleles than in mosquito adapted populations, suggesting strong selection.

In contrast to beneficial mutations, which were largely host and replicate specific, deleterious and lethal mutations exhibited significant overlap between the two hosts (*Figure 3h* and *Figure 3—figure supplement 1d*). This indicates viral protein and RNA structures and functions share common constraints in the two host environments. These constraints were further examined by evaluating how specific mutation types contribute to the viral fitness landscape (*Figure 3i–k*). As expected, synonymous mutations tended to be more neutral than non-synonymous mutations, which exhibited a bimodal distribution of fitness effects (*Figure 3k*). To obtain insights into biophysical constraints, we partitioned non-synonymous mutations into conservative substitutions (*Figure 3i–k*, 'Cons.'), which

do not significantly change the chemical and structural properties of sidechains, and non-conservative which do (*Figure 3i–k*, 'Non-cons.')(*Pechmann and Frydman, 2014*). Non-conservative changes exhibited significantly greater deleterious fitness effects than conservative changes, emphasizing the impact of biophysical properties on fitness effects as well as the sensitivity of our approach to uncover these differences (*Pechmann and Frydman, 2014*). As expected, lethal alleles were enriched in nonsense mutations (*Figure 3j*, 'Stop') as well as nonsynonymous substitutions (*Figure 3j*). These findings reveal the structural biophysical constraints shaping the DENV adaptive landscape and constraining viral diversity.

## Linking population composition to experimental phenotypes

Examining allele frequency in the populations over passage revealed a shift in the distribution of allele fitness over the adaptation experiment, which reflects the rate at which new mutations flow into the population and the strength of selection acting on those mutations (*Figure 4a*; *Figure 4—figure supplement 1a*). In early passages, the population is dominated by neutral and deleterious alleles that arise continuously in each replication cycle (*Figure 3*). In later passages, when rare beneficial mutations begin to accumulate under positive selection, we observe a concurrent loss of deleterious and neutral mutations, likely driven out by negative selection in a soft selective sweep. However, because most mutations arising during replication are deleterious or neutral (*Figure 3f*), the deleterious genetic load is never fully purged from the viral populations.

The Fundamental Theorem of Natural Selection dictates that the mean relative fitness of a population should increase during adaptation (*Kimura, 1958*; *Orr, 2009*). Given the low probability of acquiring multiple mutations per genome per replication cycle (*Figure 3f*), individual alleles were treated as independent of each other in our previous analysis of fitness. However, linking the fitness of individual alleles to the dynamics of genomes, and estimating the aggregate effect of individual mutations, requires the estimation of haplotypes. To this end, we used the estimates of mutational fitness effects and frequency trajectories of individual alleles to estimate the aggregate changes in fitness in the populations over the course of passage (*Figure 4b*). For each population, we generated a collection of reconstructed haplotypes by sampling from our empirical frequencies. We then estimated the corresponding genomic fitness values, $W$, as the product of the fitness effects across all sites in the reconstructed genome (*Figures 3*, *4b*). As expected, the median $W$ of these simulated populations increased throughout passage in a given cell type (*Figure 4c*), consistent with the dynamics of the individual constituent alleles (*Figure 4a*).

Next, we compared the genotype-based median fitness of the population (*Figure 4d*) with the experimental phenotype observed in the corresponding viral population (*Figure 1*). We chose mean absolute viral titers (*Figure 1c*) as a gross measure of population replicative fitness and adaptation to the host cell. We observed a striking correlation between viral titers and the calculated median $W$ of each population, based on allele frequency trajectories alone (*Figure 4d*; R values ranging from 0.45 to 0.98). This correlation suggests that the comprehensive measurement of allele frequencies can capture the phenotypic dynamics in experimental populations.

We next estimated the contribution of beneficial, lethal, and deleterious mutations to mean population fitness. To this end, we calculated the genome fitness, $W$, for each experimental lineage as described above, but taking into account only those sites with beneficial, deleterious, or lethal alleles. We compared these class-specific trajectories to the overall mean population fitness (*Figure 4e*, broken gray line) to understand how the aggregate fitness of the population reflects the balance of beneficial and deleterious mutations (*Figure 4e*). Beneficial mutations, although occurring relatively rarely, rapidly accumulate and drive the increase in the mean relative fitness (*Figure 4e*, yellow line). In contrast, deleterious alleles, which individually are present at low frequencies but occur on 40–50% of genomes, contribute a significant mutational load across passages (*Figure 4e*, purple line). The result is mean fitness of the population is less than 1.0 (parental, WT fitness) early in passage, when the deleterious load overwhelms rare beneficial mutations. Only after 4–5 passages do beneficial mutations drive the mean fitness above 1.0. Although beneficial mutations sweep in, they do not completely drive out the deleterious load. Instead, deleterious alleles reduce the mean fitness by approximately 50% across all passages (*Figure 4e*, purple line). Of note, lethal mutations (*Figure 4e*, black line) exert minimal effect on the population because they are rapidly purged and remain only at very low frequencies (at or below the mutation rate). Together, these analyses reveal how the phenotypes of large viral populations, characterized by high mutation rates, reflect the

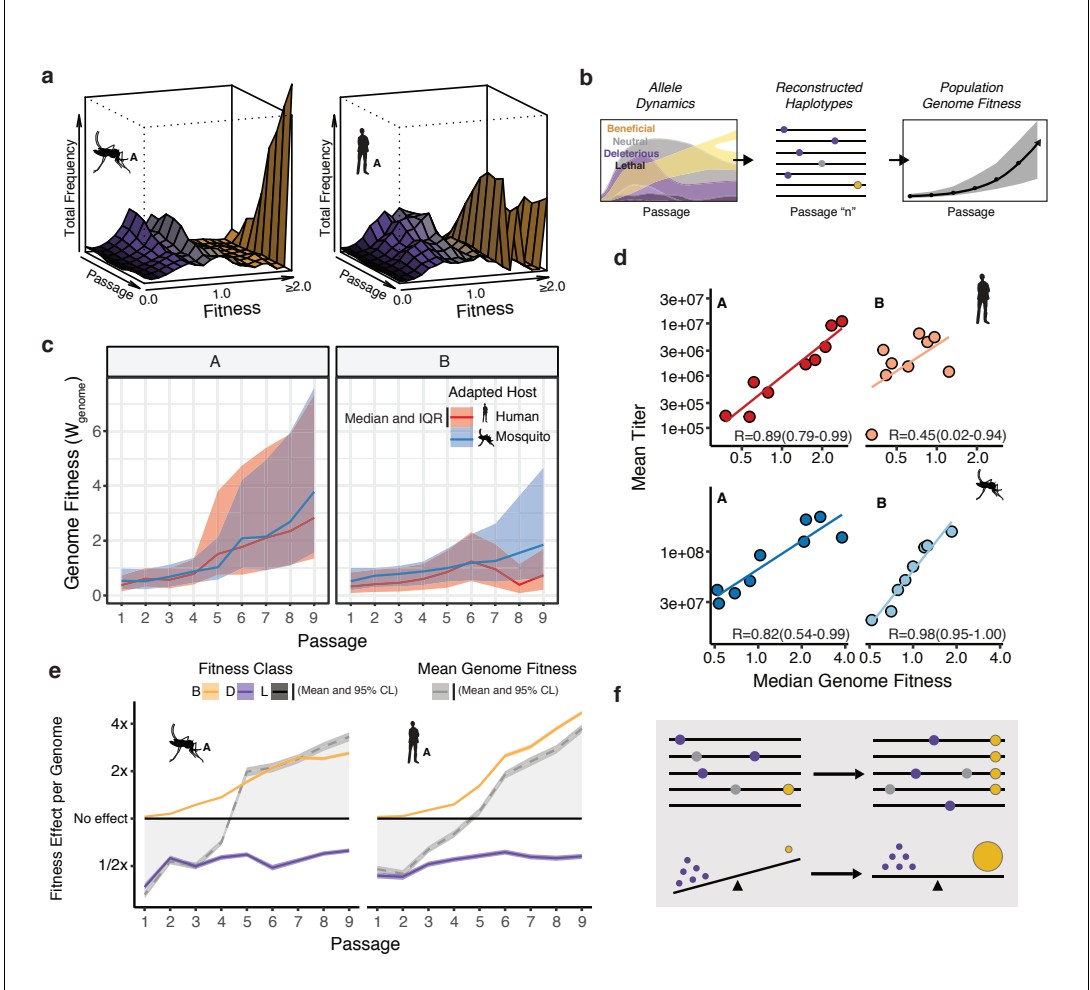

**Figure 4.** Connecting global evolutionary dynamics and population fitness. (a) Surface plot of the 'fitness wave' illustrating the change in frequency of alleles, colored by fitness bin, throughout a passage experiment. The height of the surface represents the sum of frequencies of alleles in a given bin. Deleterious and neutral mutations (purple and gray regions) make up a large proportion of the population early in the experiment. They are largely, but not entirely, driven out in later passages as beneficial mutations (yellow) increase in frequency. (b) The shift in allele fitness effects circulating in the population suggests a net increase in the fitness of genotypes within the adapting populations. To estimate the population-level change in fitness during adaptation, we used the allele frequencies in each passage (left panel) to reconstruct haplotypes for that passage (e.g. passage 'n', middle panel). The potential genotypes from each sequenced population were inferred based on empirical allele frequencies to determine the probability of a sequence identity at each site for each estimated genotype. We then computed the fitness of the resulting genotype as the product of the fitness effects across the genome. Sampling a large population of representative genotypes from the populations, we were able to generate a distribution of genome fitness values for each sequenced population (right panel). (c) Median and interquartile range of genotypic fitness ($W$) of 50,000 reconstructed genotypes sampled from the empirical allele frequencies in each sequenced population. (d) Correlation plots comparing the median genotypic fitness ($W$) of the reconstructed populations versus mean virus titer from focus forming assays (N=4). (e) Line plots showing the mean effect of beneficial (yellow), deleterious (purple), and lethal (black) mutations on mean fitness of the population (gray line). The shaded area represents the 95% confidence interval of the mean from 50,000 reconstructed genomes.

The online version of this article includes the following figure supplement(s) for figure 4:

**Figure supplement 1.** Changes in allele composition.

balance of beneficial and deleterious mutations (*Figure 4f*). In the future, using population sequencing to characterize mutational burden in different viral species will allow us to better understand how mutational tolerance and constraint on viral genomes relates to viral emergence, transmission, and long- and short-term evolution of viral populations.

## Molecular and structural determinants of dengue host adaptation

Analysis of the regions of the viral genome under positive and negative selection in each host provided insights into the molecular determinants of DENV adaptation. We calculated the mean fitness effect of non-synonymous and noncoding mutations in 21 nucleotide windows and mapped them onto the genome (*Figure 5a*). Regions of evolutionary constraint, denoted by deleterious (purple) and lethal (black) mean fitness effects, were found throughout the genome, distributed similarly between the two hosts. These likely reflect general constraints on protein structure and function. For instance, regions in non-structural proteins NS3, NS4B, NS5, and in the UTRs shared regions of strong negative selection in both hosts, which may denote key structural and functional elements. In contrast, the patterns of positive selection along the genome were different between the two hosts (yellow points in *Figure 5a*). Notably, many beneficial mutations were clustered at a few specific locations in the genome (yellow points in *Figure 5a*), suggesting adaptation relies on hotspots of host-specific selection.

To further analyze these 'hotspots' of adaptation we mapped the allele fitness values on the three-dimensional structure of dengue protein E, a well-studied transmembrane protein which forms the outermost layer of the viral envelope (*Figure 5b and c*; *Kuhn et al., 2002*). The clusters of adaptive mutations identified in mosquito cells were under negative selection in human-adapted populations (*Figure 5b*). To obtain molecular insight into the mechanisms of host-specific adaptation, we examined in more detail the loop surrounding the glycosylation site at N153, which has been extensively analyzed in previous studies (*Bryant et al., 2007*; *Hacker et al., 2009*; *Lee et al., 2010*; *Mondotte et al., 2007*) Closer examination of this loop (E152-155) (*Figure 5c*, zoomed region) revealed two dominant mosquito-adapted alleles, N153D and T155I, which lead to identical phenotypic consequences, namely to abrogate N153 glycosylation. N153D eliminates the asparagine that becomes glycosylated, while T155I disrupts the binding of the oligosaccharyltransferase mediating glycosylation (*Figure 5d*). Thus, both positively selected mosquito alleles disrupt NxT glycosylation at this site (*Chung et al., 2017*), indicating that eliminating this glycan moiety is beneficial in mosquito cells but not in human cells (*Figure 5d*). Strikingly, these findings are consistent with previous mutagenesis studies of DENV protein E, showing that losing N153 glycosylation increases DENV infectivity but impairs release of E protein in mammalian cells (*Lee et al., 2010*). Interestingly, glycosylation pathways diverge significantly between humans and insects, yielding very different final glycan structures (*Yap et al., 2017*). Since this loop is a primary site of structural variation in E proteins of dengue and related flaviviruses, including Zika virus (*Sirohi et al., 2016*), its diversification may reflect past cycles of host-specific selection acting on this region of E. The congruence of these previous mutagenesis analyses and our findings highlight the power of our approach to reveal new molecular determinants of DENV adaptation.

A major roadblock in antiviral development is the ability of viruses to mutate binding sites for antiviral drugs (*Richman, 2006*). Because CirSeq can detect alleles at frequencies at or below to the mutation rate, it permits detection and quantification of negative selection, revealing sites that are critical for viral replication. Therapies targeting these highly constrained regions under strong negative selection may be less susceptible to resistance mutations. For instance, both the RNA polymerase and the methyltransferase active sites of NS5 are enriched in lethal mutations in residues contacting the enzyme substrates (*Figure 5f*). Further analysis of mutations in the methyltransferase residues contacting its ligands SAM and mRNA cap illustrates the ability of our approach to delineate between subtle fitness differences. We find residues that contact the ligands through sidechain interactions are under strong negative selection, with most mutations highly deleterious. In contrast, residues that interact with the ligands through backbone interactions were relaxed in their fitness effects (*Figure 5g*). Thus, such high-resolution evolutionary analyses could complement structure-based antiviral drug design by identifying regions of reduced evolutionary flexibility, which may be less prone to mutate to produce resistance.

Our analyses also captured key differences in the evolutionary constraints on the viral 3′ UTR (in *Figure 5h*). We find that stem-loop II and the nearly identical stem-loop I in the 3′ UTR show significant shifts in mutational fitness effects between human and mosquito cells (*Figure 5h and i*). These stem-loops are conserved across flaviviruses and form a 'true RNA knot,' capable of resisting degradation by the exonuclease XRN1 (*Akiyama et al., 2016*; *Chapman et al., 2014a*, *Chapman et al., 2014b*). Comparing our results with previous analyses studies of the 3′ UTR further reveals how host-

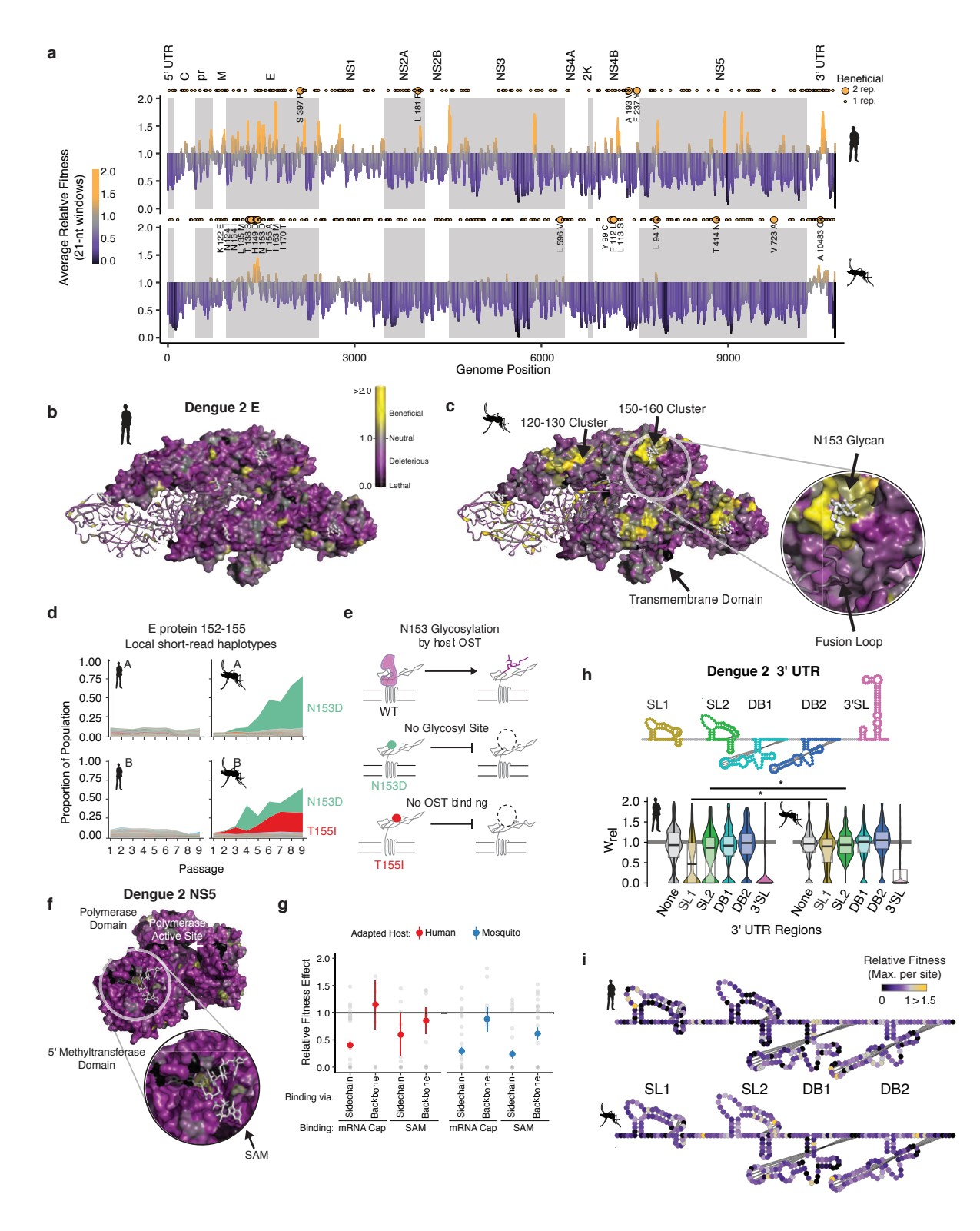

**Figure 5.** Structural analysis reveals hotspots of viral adaptation. (**a**) Bar plot of the mean fitness effect of alleles in 21 nt windows across the DENV genome. Fitness estimates from both replicates are used to compute means. Synonymous alleles are removed to emphasize the fitness effects of coding changes. Yellow points above each line denote the locations of beneficial mutations (95% CI >1). Larger, labeled yellow points denote beneficial mutations identified in both replicates of host cell passage. (**b**) The empirical fitness estimates displayed on a trimer of envelope and M

*Figure 5 continued on next page*

*Figure 5 continued*

proteins (PDB: 3J27) in an antiparallel arrangement, similar to that found on the mature virion (*Zhang et al., 2013*). To emphasize rare sites of positive selection, the color of each residue represents the maximum of the lower confidence intervals of fitness effect estimates at that site. Human-adapted populations show significant negative selection on the envelope protein surface, with no residues showing significant positive selection (yellow color). (c) Mosquito-adapted DENV exhibit two patches of pronounced positive selection on the exterior face of the virion (labeled 120–130 and 150–160). These clusters are absent from the human-adapted populations. Cluster *150–160* (Zoom), consists of a loop containing a glycosylation site at N153. This loop and glycan enclose the viral fusion loop of the anti-parallel monomer prior to activation and rearrangement in the endosome after entry. (d) Plots of the frequency of local read haplotypes for the area overlapping N153 and T155. Mutations at N153 (N153D) and T155 (T155I) are positively selected in mosquitoes, but never occur together on individual reads. (e) Schematic describing the phenotypically equivalent effects of the N153D and T155I mutations. These mutations block recognition and modification by the host oligosaccharyltransferase (OST), which initiates glycosylation. (f) CirSeq also reveals patterns of negative selection. Patches of significant evolutionary constraint can be seen around the methyltransferase active site highlighted by numerous positions with lethal fitness effects (Zoom). (g) Comparison of fitness effects of non-synonymous mutations targeting residues in NS5-MT that interact with the 5 (h) Insights into host-specific RNA structural constraints. Violin plot comparing the fitness effects of mutations in the stem-loop (SL) and dumb-bell (DB) structures of the 3′ UTR RNA of DENV2 shown in the schematic. Fitness effects of mutations in the conserved structures reveal differences in fitness effects associated with SLI and SLII in human- and mosquito-adapted dengue virus populations. (i) Nucleotide-resolution map of fitness effects on the viral 3′ UTR reveals regions of SL1 and 2 that are under tighter constraints in human passage.

The online version of this article includes the following figure supplement(s) for figure 5:

**Figure supplement 1.** Panels showing the surface and interior views of the DENV E and M proteins.

adaptation can overcome an environmental challenge through different solutions. Thus, previous studies showed DENV adapts to mosquito through deletions in stem-loops I and II (*Villordo et al., 2015*). Our analyses reveal point mutations disrupting the structure of these loops are also beneficial in mosquitoes, highlighting the diversity of stem-loop altering mutations available to increase fitness in specific environments. Recently, a study reporting passage of DENV1 in *Ae. albopictus* mosquitoes found identical mutations altering SLII stability (*Bellone et al., 2020*). These examples illustrate our ability to recapitulate and identify subtle shifts in the DMFE to uncover molecular mechanisms of selection and adaptation operating on DENV populations in cells and in host populations.

## Defining biophysical principles of dengue virus evolvability

The clusters of adaptive mutations in specific regions of the dengue genome suggest discrete elements targeted by selection in each host. We next examined the structural and functional properties of these elements to better understand the biophysical properties that influence DENV host adaptation.

The dengue polyprotein consists of soluble and transmembrane domains. We found that transmembrane domains were depleted of beneficial mutations and enriched in lethal mutations (*Figure 6a*). Thus, despite differences in lipid composition of human and insect membranes (*Hafer et al., 2009*; *Opekarová and Tanner, 2003*), the transmembrane regions of DENV disfavor changes during host cell adaptation. For non-transmembrane DENV regions, we found striking differences between structured domains and intrinsically disordered regions (IDRs) (*Figure 6b*). Beneficial mutations were highly enriched in IDRs, but not in structured regions (*Figure 6b*). In contrast, lethal mutations were enriched in ordered domains, while strongly depleted from IDRs, highlighting the evolutionary constraints imposed by maintaining protein stability and function.

We next examined whether the patterns observed in our short-term experimental evolution (*Figure 6c*). Sequence alignments of all four major DENV serotypes were used to classify amino acid residues that are invariant across the four serotypes, those with conservative substitutions that maintain chemical properties, and those with highly variable non-conservative substitutions. Strikingly, when compared to the fitness classes derived in our study, the occurrence of lethal, detrimental, and beneficial mutations mirrored the evolutionary conservation and variance across DENV serotypes (*Figure 6c*). For instance, beneficial mutations in our study were strongly enriched in the regions of highest variation across DENV serotypes, while lethal mutations were enriched in residues that are invariant during evolution. These conclusions were supported when extending this analysis to include conservation between DENV and Zika virus (*Figure 6d*). Displaying agreement between the two compared evolutionary scales, we find that regions displaying higher constraints in long-term evolution are depleted of beneficial mutations and enriched in lethal mutations arising from our short-term cell culture analysis. In contrast, regions of higher variation between viral species have fewer

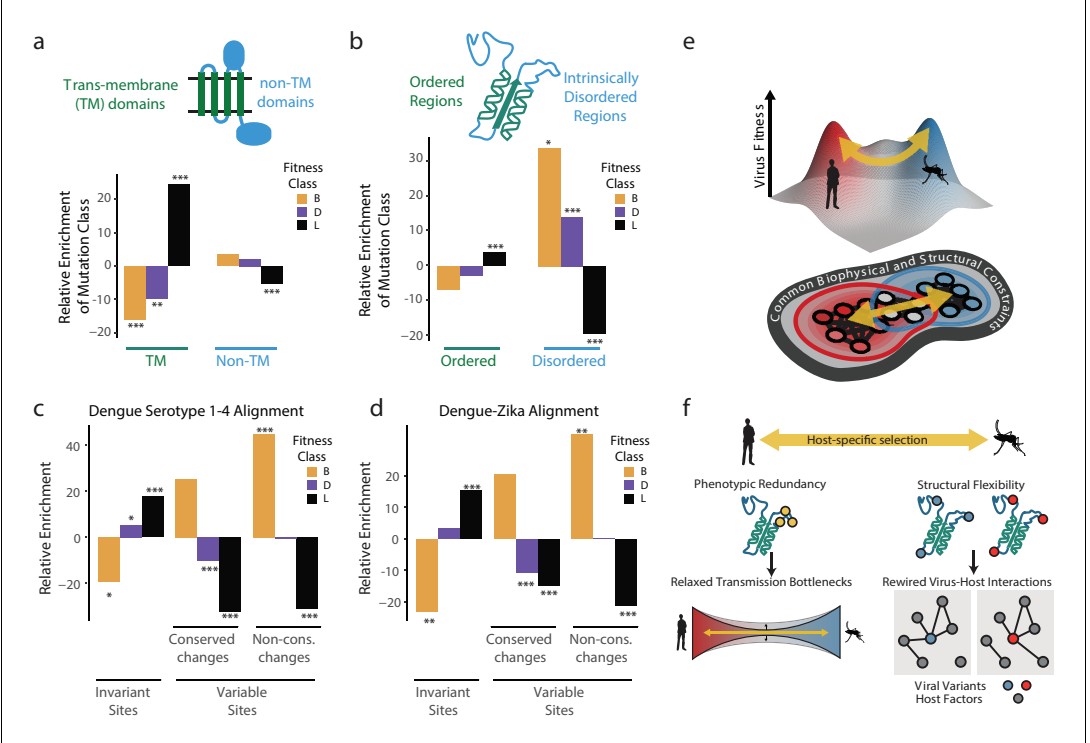

**Figure 6.** Biophysical and biological themes in DENV adaptation. (**a, b**) Distribution of mutations in regions with different biophysical characteristics. (**a**) The relative enrichment of each type of mutations (beneficial, deleterious, and lethal) in transmembrane (TM) regions versus non-TM regions (*Figure 2a*) and disordered versus ordered regions (*Figure 2b*). Relative enrichment is computed as the normalized difference in occurrence of this type of mutation in the specific region tested, and its occurrence across the entire polyprotein. Significance values (FDR-corrected, Fisher exact values) are shown (* = p<0.05, ** = p<0.01, *** = p<0.001). (**c**) Cross-dengue conservation. Distribution of mutations in regions with different levels of conservation across dengue virus strains. The relative enrichment of each type of mutations (beneficial, deleterious, and lethal) in residues that are identical ('Invariant'), similar ('Conserved') or dissimilar ('Variable') across the four dengue strains. Relative enrichment was computed as the normalized difference in occurrence of this type of mutation in the specific region tested, and its occurrence across the entire polyprotein. Significance values (FDR-corrected, Fisher exact values) are shown (* = p<0.05, ** = p<0.01, *** = p<0.001). (**d**) Zika-Dengue conservation. Distribution of mutations in regions with different levels of conservation between DENV and ZIKV virus. The relative enrichment of each type of mutations (beneficial, deleterious, and lethal) in residues that are identical, similar, or dissimilar between the two viruses. Relative enrichment was calculated as the normalized difference in occurrence fraction of this type of mutation in this specific region and its occurrence across the entire polyprotein. Significance values (FDR-corrected, Fisher exact values) are shown (* = p<0.05, ** = p<0.01, *** = p<0.001). (**e**) Visualization of a simplified landscape of dengue host adaptation. The landscape is shaped by common biophysical and functional constraints that operate similarly in both hosts, defining the outline of the fitness landscape. Positive selection of host-specific phenotypes drives host adaptation. (**f**) Host adaptation is associated with trade-offs that form a bottleneck to transmission. This bottleneck is relaxed by phenotypic redundancy and structural flexibility at key hotspots of adaptation.

The online version of this article includes the following source data for figure 6:

**Source data 1.** Pooled count data used for computing Fisher's exact test enrichment.

**Source data 2.** Fitness values and classifications used in analysis in *Figure 6*.

lethal mutations and a higher level of beneficial mutations. Thus, our simple cell culture paradigm uncovers patterns of conservation and adaptation that reflect design principles of flaviviruses cycling between human and mosquito hosts.

Together, these analyses begin to map the topography of DENV sequence space and suggest how the genomes of flaviviruses are positioned within this space to facilitate access to fitness peaks in its alternative host cell environments (*Figure 6e*). We find that a large fraction of the DENV genome sequence space does not respond to host-specific pressures. Transmembrane and structured domains are not subject to optimization through host-specific beneficial mutations, indicating these regions reside at a trade-off point for efficient replication in both hosts. Interestingly, adaptation to each host cell operates primarily through variation in flexible, surface-exposed disordered regions. IDRs have few structural constraints and tend to mediate protein-protein and protein-RNA

interactions, making them well suited for the evolutionary remodeling of virus-host-specific networks (*Charon et al., 2018*; *Goh et al., 2016*).

## Discussion

Here, we used high-resolution sequencing to quantify the contribution of beneficial and deleterious mutation in shaping the evolutionary paths of DENV populations responding to host cell-specific selective pressures. Our analysis shows that DENV populations acquire host-specific population structures defined by distinct genotype-fitness landscapes (*Figures 1* and *2*), which collectively shift the population in sequence space and result in a concurrent increase in phenotypic fitness, as assessed by focus morphology and absolute titers (*Figure 1*).

Strikingly, we find that simple models of mean population fitness derived from allele frequency measurements alone can predict phenotypic adaptation (*Figure 4*). This suggests the phenotype and evolutionary dynamics of a virus can be described by the fitness contributions of all alleles in the population. An insight of these analyses is that viral populations carry a large burden of detrimental mutations that imposes a significant fitness cost on the population that persists across passage. Previous studies showed that the high mutation load of RNA viruses is a key determinant on viral evolution and emergence (*Agrawal and Whitlock, 2012*; *Pybus et al., 2007*). Our findings quantify the cost of deleterious mutation across the dengue genome, highlighting how negative selection shapes diversity in viral populations. It will be interesting to understand how specific functional and structural features shape these patterns and contribute to overall population fitness. The ability to map evolutionary constraint across the genome allows us to highlight potential vulnerabilities that could be harnessed to develop antiviral drugs and vaccines that are refractory to the emergence of resistance and escape.

We find that adaptive mutations, including changes in coding and noncoding regions, cluster in specific regions across the DENV genome. Examination of mosquito specific alterations in a glycosylation site in protein E and in the 3′ UTR suggest that mutations in these clusters can lead to similar phenotypic outcomes. For instance, mutations that cluster in the 3′ UTR and disrupt the structure of stem-loop II, and mutations along loop 150–160 in DENV E protein that disrupt glycosylation in mosquito cells, both sites for gate-keeper mutations for mosquito transmission (*Filomatori et al., 2017*; *Mondotte et al., 2007*; *Villordo et al., 2015*). This suggests that the process of alternating host adaptation relies on maintaining access adaptive phenotypes through highly connected genetic networks made up of phenotypically redundant mutations. We propose that the phenotypic redundancy of such mutations increases the mutational target size associated with key transitions necessary for adaptation, thereby partially relieving possible bottlenecks associated with transmission and early adaptation (*Besnard et al., 2020*; *Girgis et al., 2012*).

Our study highlights the crucial role of structural constraints in shaping DENV evolution. Adaptive mutations are largely excluded from transmembrane domains and structured regions in DENV proteins. Thus, structural integrity places significant constraints on variation within these regions. It is tempting to speculate that the sequence of these arboviral domains is poised at a compromise that optimizes function in the distinct environments of human and mosquito cells (*Bourg et al., 2019*; *Shoval et al., 2012*; *Tendler et al., 2015*). Of note, our identification of highly constrained DENV regions, where most mutations are lethal, may uncover attractive targets for antivirals.

Beneficial mutations are enriched in flexible loops and intrinsically disordered regions of the DENV polyprotein (*Figure 6b*). The relaxed structural constraints of IDRs allow them to explore more mutational diversity without compromising protein folding or stability, thus enabling access to more extensive sets of adaptive mutations (*Charon et al., 2018*; *Gitlin et al., 2014*). Such plasticity may allow viral IDRs to rewire viral protein interactions with host factors, thereby driving adaptation to changing environments (*Figure 6f*). In the future, it will be informative to characterize how constraint across the genome influences the transmission of arboviruses, by restricting or enabling exploration of the genetic neighborhood and the persistence of specific subpopulations during transmission.

Notably, the link between structural properties and fitness effects measured in our study mirrors sequence conservation and variation across natural isolates of DENV and ZIKV (*Figure 6c,d*). This indicates that the relationships between adaptability, structural flexibility, and phenotypic redundancy uncovered here in a model of DENV adaptation to cultured human and mosquito cells can

suggest general principles of flavivirus evolution broadly. While arboviruses that cycle between human and mosquito represent a more extreme case of host switching, most emerging viruses must adapt to changing environments during zoonotic transmission or intra-host spreading. We propose that our simple experimental approach can map the mutational neighborhoods of viral genomes and how selection acts on specific sites of the viral genome and proteome to shape evolutionary outcomes linked to diversification, tropism, and spread for a wide range of RNA viruses and may be particularly useful to study virus without easily accessible animal models, or tools for engineering mutational screens.

# Materials and methods

### Key resources table

| Reagent type (species) or resource | Designation | Source or reference | Identifiers | Additional information |
|---|---|---|---|---|
| Strain, strain background (Dengue Virus) | Dengue Virus, Serotype 2, Thailand, 16681 | PMID:9143286 | Plasmid pD2/IC-30P | |
| Cell Line (Homo sapiens) | HepG2 | PMID:233137 | RRID:CVCL_0027 | |
| Cell Line (Aedes albopictus) | C6/36 | PMID:690610 | RRID:CVCL_Z230 | |
| Cell line (Homo sapiens) | Huh7 | PMID:30894373 | RRID:CVCL_7927 | |
| Cell Line (Homo sapiens) | Huh7.5.1 | PMID:15939869 | RRID:CVCL_E049 | |
| Cell Line (Chlorocebus sabaeus) | Vero cells | ISSN: 0047–1852 | RRID:CVCL_Z230 | |
| Antibody (Mouse anti-DENV Envelope) | Anti-E antibody | Genetex | GTX127277 | |

## Cells

Huh7 (RRID: CVCL_7927), Huh7.5.1 (RRID: CVCL_E049), HepG2 (RRID: CVCL_0027), Vero cells (RRID: CVCL_0059) were cultivated at 37°C and C6/36 cells (RRID: CVCL_Z230) at 32°C, respectively, as described previously (*Taguwa et al., 2015*). Cells lines were obtained from ATCC, validated, and tested for mycobacterium contamination.

## Viruses

DENV2 strain 16681 viral RNA was transcribed in vitro from Xba I-digested pD2/IC-30P- using the MEGAscript T7 kit (Applied Biosystems) according to the manufacturer's protocol. DENV-2 clone 16681 was isolated from a patient in Bangkok, Thailand in 1964 (*Kinney et al., 1997*), passaged in BS-C-1 (Grivet monkey) cells, six times in rhesus Macaques LLC-MK$_2$ (CCL-7) cells, in a rhesus macaque, and twice in *Toxorynchites amboinensis* mosquitoes. It was then passaged in primary Green Monkey cells, twice in LLC-MK$_2$ cells, and four times in *Aedes albopictus* c6/36 cells prior to subcloning. (*Kinney et al., 1997*).

One µg of the infectious RNA was electroporated by Gene Pulser (Bio-Rad, Hercules, CA) into Huh7 cells at 4 million cells/0.4 ml, or 10 µg infectious RNA was transfected into C6/36 at same cell number (as previously described in *Taguwa et al., 2015*). The supernatant from transfected Huh7 and C6/36 were harvested at 4 and 7 days post-electroporation, respectively. These two human and mosquito 'passage 0' populations were used to inoculate each replicate lineage on the same host cell. At each passage, virus titers in the supernatant were measured by focus-forming assay in the passaging line and adjusted to $5 \times 10^5$ FFU of DENV for the next passage onto one 10 cm dish containing $5 \times 10^6$ of Huh7 or C6/36 cells at an MOI of approximately 0.1. The culture medium was collected before the cells showed a severe cytopathic effect (CPE). In C6/36 cells, virus was collected at 72 hpi, in Huh7, due to a shift in the phenotype of the adapted lines, both replica were collected at 48 hpi after passage 3.

## Focus-forming assay

Semi-confluent cells cultured in 48-well plates were infected with a limiting 10-fold dilution series of virus, and the cells overlaid with culture medium supplemented with 0.8% methylcellulose and 2%

FBS. At 3 (Huh7) or 4 (C6/36) days post-infection, the cells were fixed by 4% paraformaldehyde-in-PBS, stained with anti-E antibody and visualized with a VECTASTAIN Elite ABC anti-mouse IgG kit with a VIP substrate (Vector Laboratories, Burlingame, CA USA). The entire wells of 48-well plates were photographed by Nikon DSLR camera D810, and each foci size was measured by Image-J. Each experiment was performed in duplicate.

## Quantitative real-time PCR (qRT-PCR)

The intracellular RNAs were prepared by phenol-chloroform extraction. cDNA was synthesized from purified RNA using the High-Capacity cDNA Reverse Transcription Kit (Life Technologies), and qRT-PCR analysis performed using gene-specific primers (iTaq Universal Supermixes or SYBR-Green, Bio-Rad) according to manufacturers' protocols. Ct values were normalized to GAPDH mRNA in human cells or 18S rRNA in mosquito cells. qRT-PCR primers are listed in Table S1. Each experiment was performed in triplicate.

## CirSeq and analysis of allele frequencies

For preparing CirSeq libraries, each passaged virus ($5 \times 10^6$ FFU) was further expanded in parental cells seeded in four 150 mm dishes. The culture medium was harvested before the appearance of severe CPE, and the cell debris was removed by centrifugation at 3000 rpm for 5 min. The virion in the supernatant was spun down by ultracentrifugation at 27,000 r.p.m, 2 hr, 4°C and viral RNA was extracted by using Trizol reagent. Each 1 µg RNA was subjected to CirSeq libraries preparation as described previously (*Acevedo and Andino, 2014*).

The CirSeq pipeline allows error control in RNAseq through consensus generation and quality filtering to overcome the intrinsic error rate associated with reverse transcription. The experimental and computational are described in detail previously (*Acevedo and Andino, 2014*). Briefly, purified viral RNA is fragmented to yield 80–100 bp fragments, circularized, and subject to rolling-circle reverse transcription. This procedure yields tandem reverse transcripts that are used to correct reverse transcription errors. Variant base-calls and allele frequencies were then determined using the CirSeq v2 package (https://andino.ucsf.edu/CirSeq). Circularized repeats are oriented to the reference genome and variants are called from raw reads based on phred33 scores of 20 (99% accuracy). These tandem variant-called reads are then aligned to each other to generate consensus sequences with a theoretical error of 1e-06. Technical replicates of passaged libraries, and individual sequencing lanes, were compared after CirSeq mapping and pooled for analysis of fitness. Raw reads are deposited at Bioproject PRJNA669406. All consensus, and mapped reads from CirSeq are deposited at https://purl.stanford.edu/gv159td5450.

## Calculation of relative fitness

An experiment of $N$ serial passages will produce, for any given single nucleotide variant (SNV) in the viral genome, a vector, $X$, of variant counts at each passage, $t$:

$$X = \{x_1, \ldots, x_t, \ldots, x_N\}$$

And, a vector Y containing the corresponding coverages at each passage, $t$:

$$Y = \{y_1, \ldots, y_t, \ldots, y_N\}$$

As explained previously (*Acevedo et al., 2014*; *Acevedo and Andino, 2014*), the relative fitness of a SNV, $w$, at time $t$ can be described by the linear model:

$$\frac{x_t}{y_t} = \frac{x_{t-1}}{y_{t-1}} * w_t + \mu_{t-1} \tag{1}$$

where $\mu_{t-1}$ is the estimated mutation rate for the variant at time $t-1$ (described previously *Acevedo and Andino, 2014*). This model requires only two consecutive passages to estimate a relative fitness parameter. However, to account for and quantify passage-to-passage noise in the estimates of relative fitness we used the values of $w$ across the first seven passages (before trajectories are strongly influenced by clonal interference) to estimate the mean and variance of $w$ for each SNV.

To account for genetic drift in our experiment, we used a similar approach as (*Acevedo et al., 2014*; *Acevedo and Andino, 2014*). At each passage, a fixed number of focus forming units, $\beta$, are

used to infect each subsequent culture. In each $\beta$ virions, $b_{t-1}$ of them will carry a given SNV. Therefore, $\frac{b_{t-1}}{\beta}$ can be used to express the frequency of that SNV in the transmitted population, which when substituted for the term, $\frac{x_{t-1}}{y_{t-1}}$, in the right side of *Equation (1)* will yield:

$$\frac{x_t}{y_t} = \frac{b_{t-1}}{\beta} * w_t + \mu_{t-1}$$

or:

$$w_t = \frac{\beta \left( \frac{x_t}{y_t} - \mu_{t-1} \right)}{b_{t-1}} \tag{2}$$

where:

$$b_{t-1} \sim B\left( \frac{x_{t-1}}{y_{t-1}}, \beta \right) \tag{3}$$

Given that $\beta$ is constant across passages ($5\times10^5$ FFU), we need only calculate $b_{t-1}$ in order to estimate $w_t$ values. Since we do not know the real value of $b_{t-1}$ for any variant, especially for low frequency variants which are sensitive to bottlenecks, we need to estimate it. This can be done by sampling $m$ times from equation (3). Such sampling is described by a Poisson distribution, then:

$$arg\,max\,\frac{\lambda^k}{k!}e^{-\lambda} \tag{4}$$

will give a maximum likelihood estimate for $\lambda = b_{t-1}$, while the upper bound of $k$ is given by $\beta$. Doing so for each $x$ from time 1 to N-1 of gives a vector, $\underline{B}$, of $b$ values: $\underline{B} = \{b_1, b_2, \ldots, b_{N-1}\}$. Finally, we estimate N-1 $w$ values by solving equation (2) using each element of $\underline{B}$. This gives a vector $\underline{W} = \{w_{t_1} \ldots, w_{t_{N-1}}\}$.

Then, the slope of the linear regression over the cumulative sum of $\underline{W}$ yields the estimated relative fitness, $w$, of a given SNV. For this regression, we employed the Thiel-Sen regression method, given that some of our vectors $\underline{W}$ contains outliers as the result of $\underline{X}$ having zeros due to poor coverage. This regression will allow for the estimate to be robust to those outliers, to avoid classifying them as detrimental variants because spurious zeros. At the same time, for $\underline{W}$ with a majority of zeros and some positive observations, that are likely to come from elements in $X$ that are not significant (i.e. sequencing errors), the Thiel-Sen estimate will give more weight to the real zero values, classifying them as lethal or deleterious, and neglecting the effect of the positive elements in $W$. Finally, we also obtain an estimate of the 95% confidence interval by the procedure described previously (*Sen, 1968*) and implemented in the 'deming' package (*Therneau, 2014*).

## Calculation of mean fitness

To estimate the effect of the observed evolutionary dynamics on the fitness of individual viral genomes in the population in the absence of haplotypic information, we generated a population of reconstructed viral genomes sampled from our empirical allele frequencies. Although many software packages for the probabilistic reconstruction of haplotypes from deep sequencing reads are available (recently reviewed in *Eliseev et al., 2020*), they reconstruct haplotypes representing the most common genotypes and do not capture rare variants present in diverse populations. Because our intention was to estimate the aggregate influence of deleterious load on the populations, we developed a method for estimating the expected distribution of genome fitnesses from our empirical allele frequency measurements using random sampling. For each reconstructed genome, we select a sequence identity, and corresponding fitness effect estimate, $w$, at each position with a probability equal to its empirical frequency from the corresponding sequenced population. Estimates of fitness effects, $w$, for the selected alleles along the genome are used to compute the fitness of the reconstructed genome, $W$, as the product of the fitness estimates across all positions:

$$W = \prod_{i=1}^{n} w_i \tag{5}$$

A total of 50,000 genomes were reconstructed for each sequenced population to estimate the distribution of expected genome fitness values in the population; similar results were obtained with independent samples. To estimate the contribution of the individual classes of mutation to the aggregate population fitness (*Figure 4e*), a similar collection of genomes was reconstructed as described above, however, the estimated genome fitness, $W$, is computed as the product of variants of a given fitness class (beneficial, deleterious, neutral), treating others as neutral ($w_i = 1.0$) to mask their effect. The resulting estimate of genome fitness reflects the isolated influence of these variants in the population. Scripts can be found in the GitHub repository, https://github.com/ptdolan/Dolan_Taguwa_Dengue_2020; *Dolan, 2021*; copy archived at swh:1:rev:adbf0dd213c5c9b422e55a9d97aeae9e7e64279f.

## Dimension reduction of genotypic data

Principal components analysis was performed on the unscaled population allele frequencies using the 'princomp' function in the R base (*R Development Core Team, 2015*). Calculation of Reynold's Θ was performed using the adegenet (*Jombart, 2008*) and poppR (*Kamvar et al., 2014*) packages in R (*R Development Core Team, 2015*). Classical MDS (by *Torgerson, 1958*) was performed to embed the pairwise Reynolds distances (Θ) (*Reynolds et al., 1983*) between the viral populations in 2-dimensions.

## Dimension reduction of phenotypic data

Stress Minimization by Majorization (implemented in *SMACOF* [*de Leeuw and Mair, 2011*; *CRAN, 2020*]) was used for the ordination of cells and viruses based on empirical relative titer data. The input distance matrix was generated from the mean of $\log_{10}$ titer measurements for (N = 4) focus formation assays on each passaged population on each of five cell lines: Huh7, Huh7.5.1, C6/36, HepG2, and Vero. Titer values were $\log_{10}$ transformed and subtracted from the maximum $\log_{10}$(titer) for each cell line to yield a matrix of Cell-to-Population distances, where the minimum distance represents the highest relative viability for each virus population.

## Structural analysis

Fitness values for non-synonymous mutations were displayed on available dengue pdb structures using pyMol2 (Schrödinger). Data was aligned to structures using in-house scripts. Briefly, protein sequences for each chain in the PDB structure are mapped to the dengue 2 reference polyprotein sequence using the Smith-Waterman algorithm for pairwise alignment (implemented in 'SeqinR' package). To emphasize regions of positive selection, the values displayed on the structures represent the lower 95% confidence limit of the fitness estimate. Where multiple non-synonymous alleles could be mapped to a single residue, the maximum of the lower 95% confidence limits were displayed to emphasize the most significantly positively selected alleles at any position.

## Biophysical properties analyses

We have identified transmembrane regions using TMpred (*Hofmann, 1993*), taking regions with a score above 500 as bona fide transmembrane regions. For disorder prediction, we used IUPred2A (*Mészáros et al., 2018*), using the 'long' search mode with default parameters. We took residues with a value >0.4 to be disordered. We used Anchor from the same IUPred2A package, to find regions within disordered regions that likely harbor linear motifs, using the default Anchor parameters and taking residues with a score >0.4 to be part of motif-containing regions. For each of these regions (TM, non-TM, ordered, disordered, and motif-embedding disordered regions), we have computed the fraction of non-synonymous mutations that belongs to each mutation category (beneficial, neutral, deleterious, and lethal). We then compared these to the respective fractions of the four categories in non-synonymous mutations across the entire polyprotein. We used a one-sided Fisher exact test to test for enrichment (or depletion) in each of the biophysically-defined regions, in comparison with the entire polyprotein, and adjusted the p-values using the Benjamini-Hochberg (*Hochberg and Benjamini, 1990*) correction (*Figure 6—source data 1* and *Figure 6—source data 2*). We plot the relative enrichment for different categories of mutations across different biophysical regions. Relative enrichment is computed as the difference between the fraction of occurrence in the tested region and the fraction of occurrence in the entire polyprotein, divided by the occurrence

in the entire polyprotein. For example, relative enrichment of lethal mutations in TM region is calculated as: $\frac{fraction_{B_{TM}} - fraction_{B_{PP}}}{fraction_{B_{PP}}}$.

## Cross viral strain and species analysis

We have aligned and compared the conservation of each residue in the polyprotein of the dengue serotype we used (serotype 2) with DENV1, 3, and 4 serotypes using CLUSTALW (*Thompson et al., 1994*). The UNIPROT accessions for each of the four aligned polyprotein sequences used in the analysis are as follows: serotype 1, P17763 - POLG_DEN1W; serotype 2, P29990 - POLG_DEN26; serotype 3, Q6YMS4 - POLG_DEN3S; serotype 4, P09866 - POLG_DEN4D. The Zika polyprotein sequence used was - A0A024B7W1. We extracted from the multi-sequence alignment the residues that are conserved across the four serotypes and Zika (identical), residues that are substituted by a similar residue, and residues that have dissimilar substitutions or gaps. We then compared the distribution of mutations from the four categories, based on our experimental data analysis (beneficial, neutral, deleterious, and lethal mutations) with their distribution across the entire polyprotein. This was carried out as described in 'Biophysical analysis'.

## Data and code availability

All data for generating plots, scripts, and output from CirSeq (including mapped read files) have been deposited and are available at the persistent URL: https://purl.stanford.edu/gv159td5450. Data used for generating all *Figures 1–5* are found in *Supplementary file 1*. Scripts for reanalyzing the fitness data and creating all figures are deposited at: https://github.com/ptdolan/Dolan_Taguwa_Dengue_2020. Sequencing data will be released upon publication at Bioproject PRJNA669406.

## Acknowledgements

Research reported in this publication was supported by National Institutes of Health grants AI127447 (JF), AI36178, AI40085, AI091575 (RA), F32GM113483 (PTD), a DARPA Prophecy Award and fellowships from the Naito Foundation (ST) and Uehara Memorial Foundation (ST), and Grant No 2019037 from the United States-Israel Binational Science Foundation (BSF) (TH and RA). We thank the Frydman and Andino labs for discussions and Prof. Marc Feldman and Dmitri Petrov and their labs for constructive comments on the work.

## Additional information

### Funding

| Funder | Grant reference number | Author |
| --- | --- | --- |
| National Institutes of Health | AI127447 | Judith Frydman |
| Naito Foundation | | Shuhei Taguwa |
| Uehara Memorial Foundation | | Shuhei Taguwa |
| National Institutes of Health | F32GM113483 | Patrick T Dolan |
| National Institutes of Health | AI091575 | Raul Andino Judith Frydman |
| United States-Israel Binational Science Foundation | 2019037 | Tzachi Hagai Raul Andino |

The funders had no role in study design, data collection and interpretation, or the decision to submit the work for publication.

### Author contributions

Patrick T Dolan, Conceptualization, Data curation, Software, Formal analysis, Validation, Investigation, Visualization, Methodology, Writing - original draft, Writing - review and editing; Shuhei Taguwa, Investigation; Mauricio Aguilar Rangel, Data curation, Software, Formal analysis; Ashley

Acevedo, Data curation, Investigation, Methodology; Tzachi Hagai, Formal analysis; Raul Andino, Conceptualization, Resources, Software, Supervision, Funding acquisition, Methodology, Writing - original draft, Project administration, Writing - review and editing; Judith Frydman, Conceptualization, Funding acquisition, Investigation, Writing - original draft, Project administration, Writing - review and editing

### Author ORCIDs
Patrick T Dolan ⬦ https://orcid.org/0000-0002-4169-0058
Tzachi Hagai ⬦ https://orcid.org/0000-0002-4575-6624
Raul Andino ⬦ https://orcid.org/0000-0001-5503-9349
Judith Frydman ⬦ https://orcid.org/0000-0003-2302-6943

### Decision letter and Author response
Decision letter https://doi.org/10.7554/eLife.61921.sa1
Author response https://doi.org/10.7554/eLife.61921.sa2

## Additional files

### Supplementary files
• Supplementary file 1. Table of fitness estimates, confidence intervals, annotations, mutation classifications, and computational statistics for all data presented here.

• Supplementary file 2. Class-specific mutation rate and error estimates for each passaged population.

• Transparent reporting form

### Data availability
All data has been deposited and is available at the persistent URL: https://purl.stanford.edu/gv159td5450 - All code for analysis and figure generation is deposited in GitHub: https://github.com/ptdolan/Dolan_Taguwa_Dengue_2020 [copy archived at https://archive.softwareheritage.org/swh:1:rev:adbf0dd213c5c9b422e55a9d97aeae9e7e64279f/ ]. Sequencing Data has been deposited as BioProject: PRJNA669406.

The following datasets were generated:

| Author(s) | Year | Dataset title | Dataset URL | Database and Identifier |
|---|---|---|---|---|
| Dolan, PT, Taguwa, S, Aguilar Rangel, M, Acevedo, A, Hagai, T, Andino, R, Frydman, J | 2020 | Principles of dengue virus evolvability derived from genotype-fitness maps in human and mosquito cells | https://doi.org/10.1101/2020.02.05.936195 | Stanford Digital Repository, 10.1101/2020.02.05.936195 |
| Dolan PT, Taguwa S, Aguilar Rangel M, Acevedo A, Hagai T, Andino R, Frydman J | 2021 | Dengue virus single-host adaptation | https://www.ncbi.nlm.nih.gov/bioproject/PRJNA669406 | NCBI BioProject, PRJNA669406 |

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
