## [Decision Letter]

**Acceptance summary:**

This work uses experimental evolution to investigate Dengue virus adaptation to mosquito and mammalian cells. By applying population genetics principles to high-fidelity ultra-deep sequencing data, the authors have measured with unprecedented detail the fitness effects of new mutations in an arbovirus. Despite the fact that laboratory evolution represents a highly simplified system, the experimental results recapitulate certain patterns of genetic diversity found in natural Dengue and Zika populations.

**Decision letter after peer review:**

Thank you for submitting your article "Principles of dengue virus evolvability derived from genotype-fitness maps in human and mosquito cells" for consideration by *eLife*. Your article has been reviewed by three peer reviewers, and the evaluation has been overseen by a Reviewing Editor and Patricia Wittkopp as the Senior Editor. The following individual involved in review of your submission has agreed to reveal their identity: Christopher JR Illingworth (Reviewer #1).

The reviewers have discussed the reviews with one another and the Reviewing Editor has drafted this decision to help you prepare a revised submission.

Summary:

This manuscript describes in vitro studies of dengue virus fitness for replication in human and mosquito cells. Ultra-deep sequencing was used to assess the fitness effects of mutations that arose during serial passages. The findings recapitulated those of previous studies, such as host-specific adaptation when one host cell was bypassed, but also characterized beneficial and deleterious mutations in more detail with greater quantitative estimates.

The consensus opinion is that this is high-quality article, which merits publication provided that some weaknesses are appropriately addressed. After discussing this with the reviewers, we have agreed on a list of changes that we are asking you to incorporate in the revised version of the manuscript.

1) The choice of C6/36 and Huh7 cell lines for evolving DENV imposes certain limitations to the relevance of the study. C6/36 cells have a dysfunctional antiviral RNAi response [Brackney et al., 2010, Scott et al., 2010]. Considering how important RNAi is for the immune response to viral infection in insects, this makes C6/36 very permissive cells, ideal for culturing or isolation. However, it makes them a questionable choice to model the arthropod host. Aag2 or A20 cells (who are both *Aedes aegypti* cells) have better immune capacity and would have been better choices. A similar caveat applies to primate cells, all of which are highly permissive because they are deficient in one or several important innate immunity pathways.

We hence request the authors to, first, make a clearer statement about the possible cell line choices that were available for this experimental evolution study, to detail the reasons for selecting C6/36 and Huh7 cells, and to better acknowledge the limitations entailed by this choice. We understand that highly permissive and/or tumoral cells are a very common choice in experimental virology, and for this reason we don´t think this should per se justify rejection of the manuscript. Of course, many mammalian and insect specific selective factors will be present in these cells even if innate immunity is shut down (e.g. temperature, receptor usage, etc).

2) The implementation of genetic drift into the model used for estimating the selection coefficient of individual SNVs should be better explained. Importantly, new alleles can experience large frequency fluctuations even if the *N_e_* is high since, initially, the number of viral particles carrying these alleles will be small. This can produce initial stochastic loss of beneficial mutations. Please provide additional details into how this was accounted for. Please also provide an estimate of *N_e_* (maybe as the harmonic mean of actual *N*, or as *N_o_*g*, where *N_o_* is inoculum size and *g* the estimated number of generations per passage, i.e. infection cycles). This point is important because *N_e_* defines which mutations are effectively neutral versus deleterious/beneficial.

---

## [Author Response]

The consensus opinion is that this is high-quality article, which merits publication provided that some weaknesses are appropriately addressed. After discussing this with the reviewers, we have agreed on a list of changes that we are asking you to incorporate in the revised version of the manuscript.1) The choice of C6/36 and Huh7 cell lines for evolving DENV imposes certain limitations to the relevance of the study. C6/36 cells have a dysfunctional antiviral RNAi response [Brackney et al., 2010, Scott et al., 2010]. Considering how important RNAi is for the immune response to viral infection in insects, this makes C6/36 very permissive cells, ideal for culturing or isolation. However, it makes them a questionable choice to model the arthropod host. Aag2 or A20 cells (who are both Aedes aegypti cells) have better immune capacity and would have been better choices. A similar caveat applies to primate cells, all of which are highly permissive because they are deficient in one or several important innate immunity pathways.We hence request the authors to, first, make a clearer statement about the possible cell line choices that were available for this experimental evolution study, to detail the reasons for selecting C6/36 and Huh7 cells, and to better acknowledge the limitations entailed by this choice. We understand that highly permissive and/or tumoral cells are a very common choice in experimental virology, and for this reason we don´t think this should per se justify rejection of the manuscript. Of course, many mammalian and insect specific selective factors will be present in these cells even if innate immunity is shut down (e.g. temperature, receptor usage, etc).

We now justify the choice of cells and discuss the limitations of our adaptation experiment in the Results section. The choice of cell lines was driven by practical reasons; first, to ensure we could generate large enough populations of virus, and achieve the necessary depth (600,000 reads per base) to map the full spectrum of diversity and estimate fitness effects for alleles across the genome.; secondly, these cell lines are also the standard cell lines used commonly in the field. Using these common cell lines allows us to connect our results to the existing body of DENV research performed in these cells, allowing us to more fully describe these evolutionary processes to infer more mechanistic insights into how mutation and selection operate during transmission. Even with these caveats, it is remarkable that our analysis reflects evolutionary constraints observed in circulating DENV serotypes and ZIKA strains, which are naturally alternating between mosquito and humans (Figure 6).

2) The implementation of genetic drift into the model used for estimating the selection coefficient of individual SNVs should be better explained. Importantly, new alleles can experience large frequency fluctuations even if the N_e_ is high since, initially, the number of viral particles carrying these alleles will be small. This can produce initial stochastic loss of beneficial mutations. Please provide additional details into how this was accounted for. Please also provide an estimate of N_e_ (maybe as the harmonic mean of actual N, or as N_o_*g, where N_o_ is inoculum size and g the estimated number of generations per passage, i.e. infection cycles). This point is important because N_e_ defines which mutations are effectively neutral versus deleterious/beneficial.

*N_e_* is factored into the calculation of fitness effect as the *β* term (Equations 2 and 3), which represents the number of particles transferred between passages (*N_o_*; fixed to 5x10^5^ FFU after a single generation). We incorporate the uncertainty from drift using sampling from a binomial distribution to generate estimates of the true frequency of the variant. We describe these samples with a Poisson distribution, whose *λ* term (frequency of the variant) we use to generate estimates of relative fitness (i.e. the estimated change frequency between each passage) and then average over these estimates (using outlier-resistant regression) to generate a robust estimate and 95% CI of the fitness value for each substitution. See the subsection “Calculation of Relative Fitness”.